

**Reviews and syntheses: Ecological Stoichiometry of Carbon, Nitrogen,**
**and Phosphorus in Shrubs and Shrublands**
Xinru Zhang [1,6], Lin Zhang [1,5,*] , Zhong Wang [2,3], Jinniu Wang[4]
[1] State Key Laboratory of Tibetan Plateau Earth System Science, Resources and Environment (TPESRE), Institute of Tibetan
Plateau Research, Chinese Academy of Sciences, Beijing, China
[2] College of Life Sciences, Wuhan University, Wuhan 430072, China
[3] College of Science, Tibet University, Lhasa 850000, China
[4] Chengdu Institute of Biology, Chinese Academy of Science, Chengdu 610041, Sichuan, China
[5] Key Laboratory of Biological Resources and Biosafety, Institute of Plateau Biology Research, Xizang autonomous region,
Lhasa 850000, China
[6] University of Chinese Academy of Sciences, Beijing 100049, China
**\* Corresponding author**
Dr. Lin Zhang, Professor in plant ecology
Institute of Tibetan Plateau Research, Chinese Academy of Sciences,
Building 3, Courtyard 16, Lin Cui Road, Chaoyang District, Beijing 100101, China
Phone/Fax: +86 10 84097055
Email: zhanglin@itpcas.ac.cn
**Text and display items:** main text 7098 words, 173 references, 6 figures
**Article Types:** Reviews and syntheses



**Abstract:** Ecological stoichiometry examines the balance and ratios of multiple elements in ecological processes. In shrubs,
characterized by their adaptability to extreme environments such as alpine and arid, stoichiometric traits likely differ from
those in trees and grasses, reflecting unique ecological adaptations of shrubs. However, this hypothesis remains underexplored.
Here we review the state of the art of stoichiometry in shrubs and then identify research hotspots of shrub stoichiometry. Then,
we summarize the effects of climate, soil properties, phylogeny, ontogenetic differences, and human activities on stoichiometry
of shrub leaves. In addition, we compared the stoichiometry of shrublands with that of forests and grasslands. The development
process of shrubland stoichiometry research can be roughly divided into three main periods: the initial development stage
(before 2010), the fast development stage (2011-2018), and the high-quality development stage (from 2019 to the present),
with the two turning points occurring in 2011 and 2019 possibly related to the launch of major projects associated with
shrublands in China. Current studies predominantly focus on shrub leaves, with limited attention to stems, roots, and seeds.
Mean values of C, N, P, C:N, and N:P in shrub leaves globally were 454.66 mg g$^{-1}$, 18.93 mg g$^{-1}$, 1.20 mg g$^{-1}$, 23.4, and 15.8,
respectively. Shrub leaf N and P content were higher than those of trees and lower than herbs, while C content and C:N ratio
showed opposite trends. N and P content correlated positively with soil nutrients and precipitation and negatively with
temperature. Functional types also influenced stoichiometry, with deciduous and leguminous shrub species showing higher N
and P content than evergreen and non-leguminous shrubs. Overall, shrubs showed C and N content intermediate between trees
and grasses, while P content was similar across life forms. Higher N:P ratios in shrublands and grasslands suggest stronger P
limitation than in forests. Future studies should integrate above- and below-ground stoichiometry, consider phylogenetic
influences, and investigate evolutionary processes to better understand shrubland adaptation and formation mechanisms under
global change.
**Keywords:** Shrublands, ecological stoichiometry, soil, climate change, bibliometrics
**1 Introduction**
The study of plant elemental composition and ecological stoichiometry serves as a vital tool for understanding the adaptation
and evolutionary mechanisms of different species within ecosystems in response to environmental changes (Reich and Oleksyn,
2004; Tian et al., 2021). Carbon (C), nitrogen (N), and phosphorus (P) are macronutrients in plants, playing pivotal roles in
regulating primary productivity, energy flow, and material cycling (Finzi et al., 2011; Chen and Chen, 2021). Carbon provides
essential substrate and energy sources for plant physiological processes, while N and P are crucial components of plant proteins
and genetic material (Lu et al., 2023). Plants synthesize organic compounds necessary for their growth and reproduction by
metabolizing energy and taking up proportional amounts of different elements (Sterner and Elser, 2002; Ågren, 2008). Previous
studies have established that N and P are the primary limiting nutrients in terrestrial ecosystems (Koerselman and Meuleman,
1996; Tian et al., 2019a), with their abundance being crucial for regulating plant primary production and ecosystem carbon



sequestration (Ågren, 2008; Tang et al., 2018). The leaf N:P ratio serves as a key indicator for assessing whether plants are
limited by N or P (Güsewell, 2004). The theory of stoichiometric homeostasis posits that plants have the ability to maintain
relatively stable levels and ratios of C, N, and P content within their tissues across environments (Sterner and Elser, 2002;
Yang et al., 2018). The relative supply of nutrients from the soil exerts a pivotal role in regulating the C:N:P balance in plants
(Plum et al., 2015). Terrestrial plant studies at both global and regional scales have demonstrated that leaf N and P content
decrease with increasing temperature and precipitation (Han et al., 2005; Tang et al., 2018). However, some studies have also
reported that plant functional types and phylogenetic factors are the main regulators of leaf stoichiometry (He et al., 2006; Han
et al., 2011). Consequently, a deeper understanding of how environmental factors and plant traits affect the distribution of
nutrients (C, N, P) between plants and soil is warranted (Zhang et al., 2019). Exploring this issue can provide insights into the
processes and mechanisms of ecosystem nutrient cycling, contributing to the sustainable development of different ecosystems
(Chen and Chen, 2021).

Shrublands account for 14-18% of the global land cover (Broxton et al., 2014; Li et al., 2023), playing a significant role

in community succession and biological carbon sequestration (Piao et al., 2009; Liu et al., 2022). Shrublands show a
pronounced response to global changes, with their area recently expanding due to persistent climate change (Deng et al., 2021;
Wang et al., 2021). As a vegetation type dominated by mesophytic or xerophytic shrubs, shrublands show a strong adaptability
to environmental changes. Consequently, compared to forests and grasslands, the physiological characteristics and habitat
conditions of shrublands may contribute to their unique stoichiometric properties and environmental response characteristics.
Furthermore, shrubs typically exhibit moderate growth rates, prolonged lifespans, and well-developed root systems. In contrast
to rapidly growing herbaceous plants, shrubs may adopt more conservative nutrient supply strategies (Wen et al., 2021), often
thriving in environments with limited nutrient or water availability (Liu et al., 2022). However, the uniqueness of the
stoichiometry of shrublands and shrubs in comparison to forests (trees) and grasslands (herbs), as well as their responses to
environmental changes, remains unclear. Addressing these issues is crucial for understanding nutrient use strategies of shrubs
and predicting the responses of shrubland ecosystems to environmental changes especially in arid and semi-arid regions. Along
these lines, the present study first outlines the research history and current status of ecological stoichiometry knowledge in
shrublands and shrubs in China and globally. Using CiteSpace software, we conducted a comprehensive analysis of hot topics
in the field. Subsequently, the responses of shrub leaf stoichiometry to factors such as soil, climate, and phylogeny are reviewed,
as well as differences stoichiometry between shrublands (shrubs), forests (trees), and grasslands (herbs) (Fig. 1). We aim to
provide insights into the ecological adaptability of shrublands and shrubs under climate change, as well as their restoration and
management strategies.





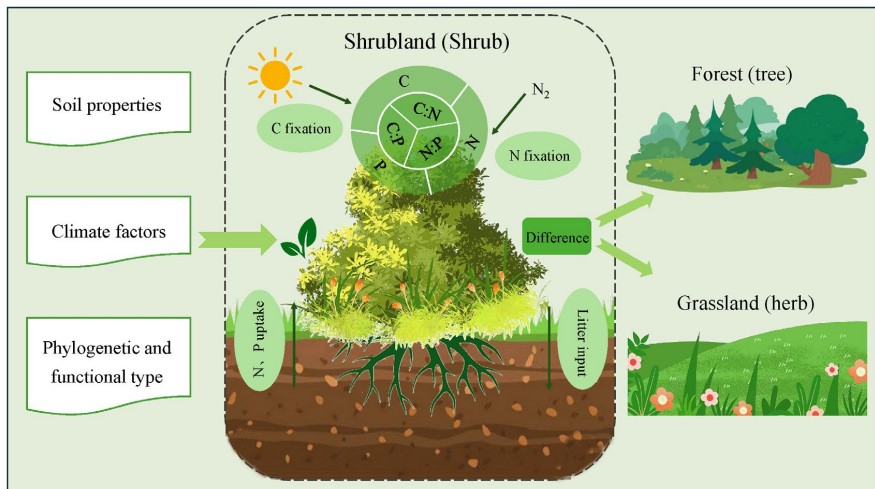

**Figure 1:** Framework diagram of C:N:P stoichiometry in shrubs and shrublands.

## 2 Data and Methods

### 2.1 Literature and Data Collection

Shrubland refers to a vegetation type dominated by shrubs, characterized by a community height typically less than 5 m and a cover greater than 30%. However, for a prolonged period, shrublands have rarely been considered as an independent entity, often being subsumed under the categories of forest or grassland, thereby neglecting its unique role. In this study, we conducted a comprehensive search encompassing both shrublands and shrubs within non-shrubland systems. Using the Web of Science Core Collection, we searched for articles published between 1997 and 2023 (as of December 31, 2023) with the keywords "shrub stoichiometr*" or "scrub stoichiometr*" or "shrub C N P" or "scrub C N P." Simultaneously, in the China National Knowledge Infrastructure (CNKI), we searched using the terms "shrubland stoichiometry" or "shrub stoichiometry." A total of 1202 articles were retrieved that met these criteria. After excluding duplicates and irrelevant publications, we got a final dataset of 540 articles, comprising 309 papers in English and 231 in Chinese. In terms of community type, 282 articles focused on shrublands, while 258 targeted shrubs within non-shrubland systems, such as shrubs in forest ecosystems.

Further screening of the selected literature was conducted as follows: (1) Five representative publications (Dong et al., 2023a; Dong et al., 2023b; Li et al., 2021; Qin et al., 2022; Tian et al., 2019) with a broad geographical scope (155.5°W~168°E, 45.28°S~68.35°N) and focusing on the stoichiometry of shrub leaves were selected. The N and P content, N:P ratios, mean annual precipitation (MAP), and mean annual temperature (MAT) were recorded to explore trends in shrub stoichiometry across large temperature and precipitation gradients. (2) Publications that targeted shrubland ecosystems and provided accessible raw data were selected. The N and P content, along with their ratios, were recorded and compared with representative





research findings from forest and grassland ecosystems. (3) Studies investigating multiple life forms within the same ecosystem
(including trees, shrubs, and herbs in forest ecosystems, as well as shrubs and herbs in shrubland ecosystems) were screened.
The mean values of N, P, and N:P ratios in leaf tissues for each life form were recorded from the literature to analyze differences
in ecological stoichiometry among different life forms within the same ecosystem. The raw data were either directly obtained
from tables in the published articles or extracted from figures using GetData Graph Digitizer software (version 2.26).
**2.2 Data Analysis**
The changes in the number of publications related to the ecological stoichiometry of shrublands (shrubs) and the proportion of
each research object were analyzed using the "ggplot2" package in R 4.2.2. Using CiteSpace visualization software (version
6.4.R1), co-word analyses were performed separately on Chinese and English literature, with the time range set from 1997 to
2023, a slice length of 1 year, and the node type selected as "Keyword." This method generated keyword clustering maps for
ecological stoichiometry studies of shrublands and shrubs, where different color blocks represent different clusters, and
concentric circles indicate the frequency of keyword occurrences. These maps were then analyzed to identify research hotspots.
Furthermore, linear regression analyses were conducted using the "ggplot2" package to examine trends in N and P content in
shrub leaves with respect to mean annual precipitation (MAP) and mean annual temperature (MAT), aiming to identify changes
in shrubs N and P content in response to climatic factors. The Wilcox test was used to test for differences in N and P
stoichiometry among forest, grassland, and shrubland ecosystems. Last, the "ggplot2" package was used to check for
differences in N and P stoichiometry between different life forms within forest and shrubland ecosystems.
**3 The Historical Development and Research Hotspots of Chemical Stoichiometry in Shrubs**
The entire developmental history can be broadly divided into three stages: the initial development stage (before 2010), the fast
development stage (2011-2018), and the high-quality development stage (2019-2023) (Fig. 2). Research on the ecological
stoichiometry of shrubs emerged only in 1997, when Castro-Díez et al. (1997) discovered that the leaf N and N:P ratios of
*Quercus coccifera*, an evergreen shrub in northeastern Spain, increased with annual precipitation, while P content showed no
significant trend. However, until 2010, only 67 publications were recorded, indicating scant attention to shrub-related research
during this period. Scholars often considered shrubs as part of forest (e.g., tall shrublands or understory shrubs) or grassland
ecosystems (companion shrubs embedded in grassland ecosystems). No Chinese articles on shrub ecological stoichiometry
were published during this time, possibly due to the late introduction of the stoichiometry concept in China. The publication
of "Ecological Stoichiometry: The Biology of Elements from Molecules to the Biosphere" by Sterner and Elser (2002) marked
the gradual maturation of the theoretical framework for plant ecological stoichiometry. During this period, the primary focus
was on the patterns of plant N-P stoichiometry and their relationships with climatic factors. Since 2011, research on shrub



ecological stoichiometry entered a rapid development stage, marked by a significant increase in the number of publications.
Sistla and Schimel (2012) reviewed the links between stoichiometry and ecosystem structure and function, proposing that
stoichiometric flexibility could serve as a regulatory factor for changes in carbon and nutrient cycling in terrestrial ecosystems,
significantly advancing research internationally. Research on this topic in China gradually emerged during this period,
potentially related to the launch of the Strategic Priority Research Program of the Chinese Academy of Sciences (Class A) in
2011, titled "Current Status, Rates, Mechanisms, and Potential of Carbon Sequestration in Shrub Ecosystems." Additionally,
the Ministry of Science and Technology of China initiated the Special Project for Basic Scientific Research in 2015, "Survey
of Shrub Plant Communities in China," which significantly promoted the steady development of research on shrub ecological
stoichiometry in China. Research began to focus on seasonal variations in shrub stoichiometry and element allocation
relationships among organs, supplementing ecological stoichiometric theories and hypotheses. For instance, Niu et al. (2013)
analysed the seasonal variation patterns of C:N:P stoichiometry in leaves of major shrub species in the Alxa Desert, finding
that leaf C and N content and C:N ratios varied little, while P, C:P, and N:P ratios showed greater variability. He et al. (2017)
discovered differences in C, N, and P content among different organs of *Sibiraea angustata* in eastern Qinghai-Tibet Plateau,
which to some extent conformed to the homeostasis theory and growth rate hypothesis (Sterner and Elser, 2002). Meanwhile,
the number of international publications did not increase significantly, and research continued to focus on the geographical
patterns of ecological stoichiometry and their climatic responses (Delgado-Baquerizo et al., 2018; Müller et al., 2017). Since
2019, the annual publication output in relevant fields has maintained above 50 papers, accounting for over 60% of the total
across three distinct periods. Notably, nearly 50% of the English articles during this timeframe originated from China,
indicating a gradual shift towards Chinese dominance in the research on shrubland ecological stoichiometry. This stage of
research primarily focused on the relationships between ecological stoichiometry, community structure, and ecosystem
function. Researchers proposed the well-coordinated elements hypothesis, which suggests that biologically coordinated
elements that regulate certain physiological functions mutually constrain each other, thus maintaining relatively stable
proportions in plants (Zuo et al., 2024). This marks a shift towards a phase of high-quality development in the related research.
Song et al. (2021) observed a significant positive correlation between the Shannon diversity index and foliar N and P in shrubs,
while a negative correlation was found with C:N and C:P ratios. Urbina et al. (2020) found that shrub encroachment
significantly increased the C content and C:N ratio in leaves and litter, while N and P content in both leaves and soil were
notably reduced. Furthermore, the establishment of the Second Tibetan Plateau Comprehensive Scientific Expedition Research
Program by the Ministry of Science and Technology of the People's Republic of China in 2019, with one of its objectives being
to promote shrubland ecosystem resource survey and management, coupled with the launch of the 2022 Xinjiang
Comprehensive Scientific Expedition Project, has significantly elevated the attention paid to the stoichiometric research of
shrubland ecosystems, particularly in desert environments.



Among the 540 selected research articles in Chinese, 34.6% focused on the C, N, P stoichiometry of a specific organ in
shrubs; most (80.75%) focused on leaves, a smaller proportion (9.10%) on roots, and yet another minority (8.56%) focused on
litter, while only a few papers encompassed branches, bark, and seeds. Out of the 540 articles, 28.5% targeted shrubland soil,
25.9% addressed both shrubs and soil, and merely a 10.9% examined the stoichiometry across multiple shrub organs (Fig.2
inset). Prior research on shrub organs primarily centered on geographical patterns of stoichiometric traits. Zhao et al. (2018)
revealed that alpine shrub leaves in river valleys had higher N and P contentand lower C content, with the latter increasing
with elevation and decreasing temperature. You et al. (2023) reported that shrub root C and N content increased with latitude,
while N content decreased with age/root diameter. Studies examining both shrubs and soil emphasized their reciprocal
influence on stoichiometry. For instance, Müller et al. (2017) observed significant positive correlations between C:N, C:P, and
N:P ratios in *Rhododendron campanulatum* and soil stoichiometry. Additionally, vegetation type has been found to impact soil
stoichiometry, with higher N and P content in soil under *Caragana korshinskii* than under *Hippophae rhamnoides*, and soil C,
N, P content and their stoichiometric ratios correlated with density of litter per square meter and root mass/volume (Wang et
al., 2022a). Research on multiple shrub organs provided deeper insights into shrub nutrient allocation strategies and
environmental adaptability. For example, Dong et al. (2023a) found that N and P content in shrubs generally followed the order
of seeds > flowers > leaves > roots > stems, indicating that shrubs allocate more nutrients to reproductive organs like seeds
and flowers to enhance reproduction efficiency. Yang et al. (2014) explored N and P allocation strategies in leaves, stems, and
roots of shrubs in northern China, revealing that N content ratios among organs exhibited allometric growth between leaves
and non-leaf organs and isometric growth among non-leaf organs, while P concentrations tended towards allometric growth
between roots and non-root organs and isometric growth among non-root organs.

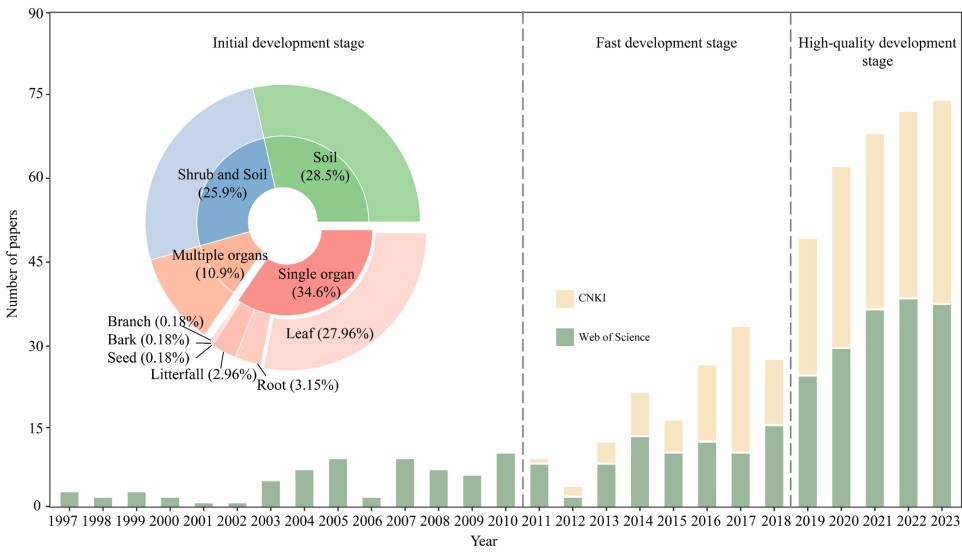


**Figure 2:** Changes in the number of papers published in the field of ecological stoichiometric studies of shrublands and shrubs from 1997
to 2023. The pie chart depicts the research subjects and their respective proportions in the literature on shrub ecological stoichiometry.



Through the analysis of the keyword clustering map (Fig. 3), we found that research on the ecological stoichiometry of
shrubs primarily focuses on eight aspects: ecological stoichiometry, foliar nutrients, drought stress, leaf morphology,
facilitation, allometric growth, community structure, and climate change. Keywords with higher frequencies include nitrogen,
carbon, growth and soil, indicating that the research mainly focused on nutrient limitations in shrub growth and the relationship
between the stoichiometry of shrubs and soil (Fig. 3). For instance, Zhang et al. (2022) explored the N and P content in leaves
of dominant shrubs under *Pinus massoniana* forests and reported that there was no significant relationship between shrub leaf
and soil C, N, P content, and their stoichiometric ratios. He et al. (2023) discovered that the leaf C content of alpine shrub
plants was higher than the global average for plants, and their growth was primarily constrained by N.

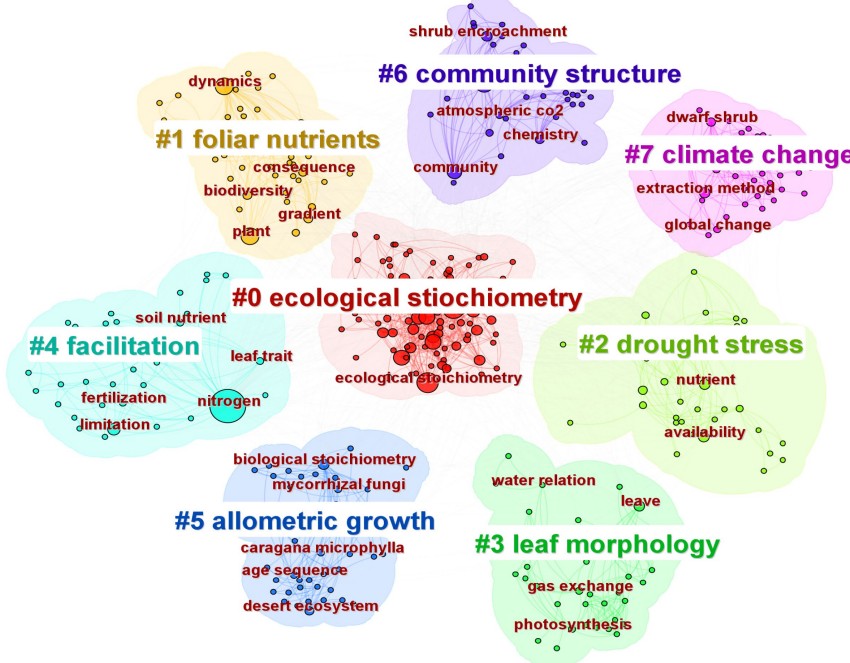


**Figure 3:** Keyword clustering map of ecological stoichiometry literature in shrublands and shrubs.
**4 Impact Factors of the C:N:P Stoichiometry in Shrub Leaves**
Based on the keyword network map (Fig. 3), apart from the high frequency of "C", "N", and "P", keywords such as "soil
nutrient", "environmental factors", "life form", "soil nutrient", and "climate change" also showed considerable frequencies
(Fig. 3), indicating that stoichiometry research focused mostly on plant leaves. Some researches have also delved into the
impacts of phylogenetic factors, fertilization (simulating nitrogen deposition), and anthropogenic activities like logging, on the
stoichiometry of shrub leaves. In summary, the attribution of leaf stoichiometry in previous studies can be broadly categorized
into three main aspects: environmental factors (encompassing soil properties and climate factors), phylogenetic factors related
to species traits, individual differences among species, and the influence of human activities like fertilization and logging.



### 4.1 Soil Properties

As the most direct source of essential nutrients for plant growth, soil exerts the most pronounced influence on plant stoichiometry. Numerous studies have found significant positive correlations between N and P content in shrub leaves and elemental content in soil (Chen et al., 2019; He et al., 2023; Lu et al., 2015; Ma et al., 2014; Xu et al., 2021; Xing et al., 2022). However, there are also reports showing no apparent relationship (Bagedeng et al., 2023; He et al., 2020; Pang et al., 2020; Zou et al., 2021), or even negative correlation between nutrient content in leaves and in soil (Dong et al., 2021; Luo et al., 2022; Xiong et al., 2022; Yang et al., 2016; Zeng et al., 2017). Possible explanations include: (1) shrub species in arid northern regions, such as *Reaumuria soongorica* and *Haloxylon ammodendron*, are drought- and low- nutrien tolerants (He et al., 2015; Yao et al., 2021; Zou et al., 2021), with low nutrient demands (Poorter, 1989), and therefore less influenced by soil nutrient content. (2) In arid and semi-arid regions, shrubs maintain high N and P content in their leaves to sustain metabolic activity, and often exhibit high nutrient resorption efficiency, thereby reducing their reliance on soil nutrients (Zou et al., 2021). In particular, leguminous shrubs can directly fix atmospheric nitrogen through symbiotic nitrogen fixation by rhizobia (Vitousek et al., 2002), leading to decoupling between plant and soil nutrient relationships (Liu et al., 2019). (3) Regulated by various environmental factors such as temperature and precipitation, the contents of elements in shrub leaves and soil may exhibit opposite trends along environmental gradients. This results in negative correlations between the nutrient contents of plants and soils (Xiong et al., 2022; Xia et al., 2023).

Apart from nutrients, other soil physical and chemical properties may influence leaf stoichiometry of shrubs. Tao et al. (2017) found that higher soil pH is detrimental to the accumulation of leaf C and N in three typical deciduous shrubs in the Gurbantunggut Desert, northwest China. He et al. (2023) reported that variations in foliar C:N:P stoichiometry of five dominant shrubs were primarily influenced by soil water content, soil bulk density and porosity in the Qilian Mountains, the northeastern Tibetan Plateau. Luo et al. (2022) reported a positive correlation between soil electrical conductivity and leaf P content of desert halophyte shrubs in Xinjiang. Many other studies have shown that soil properties such as water content, bulk density, and pH exert greater impacts on leaf stoichiometry than soil nutrients (Chao et al., 2023; He et al., 2023; Wei et al., 2021; Zhang et al., 2021). Consequently, while focusing on the influence of soil C, N, and P content on plant nutrients, it is crucial to consider soil pH, water content, bulk density, and electrical conductivity, and comprehensively evaluate the impacts of soil physical, chemical, and nutrient characteristics on the leaf stoichiometry of shrubs.

### 4.2 Climate Factors

Plants take up essential nutrients from the soil, and climate plays a pivotal role in soil development at larger scales, indirectly modulating nutrient cycling by influencing the formation, decomposition, and storage of soil organic matter (Mou et al., 2022; Ren et al., 2017), subsequently regulating ecological stoichiometry. Globally, the N and P content in terrestrial plant leaves



generally decrease with increasing annual precipitation (Reich and Oleksyn, 2004; Tang et al., 2018). We used a global dataset
comprising leaf N and P of 4,253 shrubs content along with climatic factors, spanning 582 sites and 977 species (from Tian et
al., 2019). By further integrating data from four recent publications, we found that leaf N and P content generally decreased
with rising annual precipitation, while the N:P ratio increased significantly (Fig. 4a, 4b, 4c). Similar trends were observed at
the regional scale, content such as the Loess Plateau (Zheng et al. 2007), or desert shrubs in Xinjiang Autonomous Region,
China (He et al. 2019). Nevertheless, some studies have reported contrasting trends, with shrub leaf C, N, and P content
increasing with precipitation (Dong et al., 2023b; Guo et al., 2021; Liu et al., 2013; Luo et al., 2022; Wang and Yu, 2017; Wang
et al., 2019; Wang et al., 2022b, 2022c; Yang et al., 2019; Zhao et al., 2018). Discrepancies also exist in controlled experiments;
for example, Prieto and Querejeta (2020) recorded a significant reduction in leaf N and P content after five years of water
reduction in a Mediterranean semiarid shrubland, whereas Umair et al. (2020) found no changes in leaf N and P content with
increasing water availability in a degraded karst system in southwestern China. These contrasting patterns of shrub leaf C, N,
and P stoichiometry in response to precipitation may reflect their adaptive strategies to the environment. On one hand, high N
content is considered an adaptation to arid regions (Wright et al., 2005), as it facilitates increased photosynthetic rates (Wright
et al., 2003). The negative correlation between leaf P content and precipitation is primarily attributed to soil P leaching under
high moisture conditions (Chen et al., 2013; Ordoñez et al., 2009). On the other hand, shrubs are prevalent in arid and semiarid
regions, where precipitation increases alleviate water limitation, prompting a shift towards investment-oriented growth,
including elevated N to enhance photosynthesis (Liu et al., 2017; Wei et al., 2011) and increased P to accelerate growth rates
(Luo et al., 2022).

Previous studies have established that temperature exerted a significant influence on plant ecological stoichiometry,

proposing two opposing hypotheses: the Temperature-Plant Physiology Hypothesis and the Temperature-Biogeochemistry
Hypothesis (Reich and Oleksyn, 2004). The former suggests that temperature modulates plant physiological processes, leading
to higher N and P content under low temperatures, whereas the latter posits that temperature influences plant N and P
stoichiometry by altering soil N and P availability, resulting in lower leaf N and P content at lower temperatures (Reich and
Oleksyn, 2004). On a global scale, trends in terrestrial plant leaf N and P content with temperature support the Temperature-
Plant Physiology Hypothesis, indicating a decrease in leaf N and P content and an increase in N:P ratio with rising temperatures
(Kang et al., 2010; Tang et al., 2018; Yuan and Chen, 2009). An analysis of published literature data reveals similar trends in
global leaf N and P content of shrubs, with a significant negative correlation with mean annual temperature (Fig. 4c, 4d). At
the regional level, most studies on shrub leaf stoichiometry have yielded similar results (He et al., 2019; Liu et al., 2013; Wang
and Yu, 2017; Xu et al., 2021; Yang et al., 2016; Yang et al., 2019; Zhang et al., 2019). Short-term controlled experiments
further show that warming reduces shrub leaf N (Prieto and Querejeta, 2020; Wu et al., 2019; Xu et al., 2024). However, some
studies indicate that temperature may alter N and P mineralization rates by influencing soil microbial activity, resulting in





lower shrub N and P content at lower temperatures, thereby establishing a positive correlation between temperature and N, P
content (Huang et al., 2021; Guo et al., 2021; Li et al., 2014; Wang et al., 2022b). Reich and Oleksyn (2004) reported a biphasic
response of global plant leaf N and P content to temperature, initially increasing and then decreasing, with an inflection point
around a mean annual temperature of 5°C. They suggested that the Temperature-Biogeochemistry Hypothesis dominates below
5°C, whereas the Temperature-Plant Physiology Hypothesis prevails above this threshold. For shrub leaf ecological
stoichiometry, some studies support the Temperature-Biogeochemistry Hypothesis in regions with temperatures above 5°C
(Huang et al., 2021; Guo et al., 2021; Li et al., 2014), but this has not been fully validated. Thus, while cases exist supporting
the Temperature-Plant Physiology Hypothesis for the influence of temperature on shrub leaf stoichiometry, differences may
arise due to variations in study species and regions, with the underlying reasons remaining unclear.

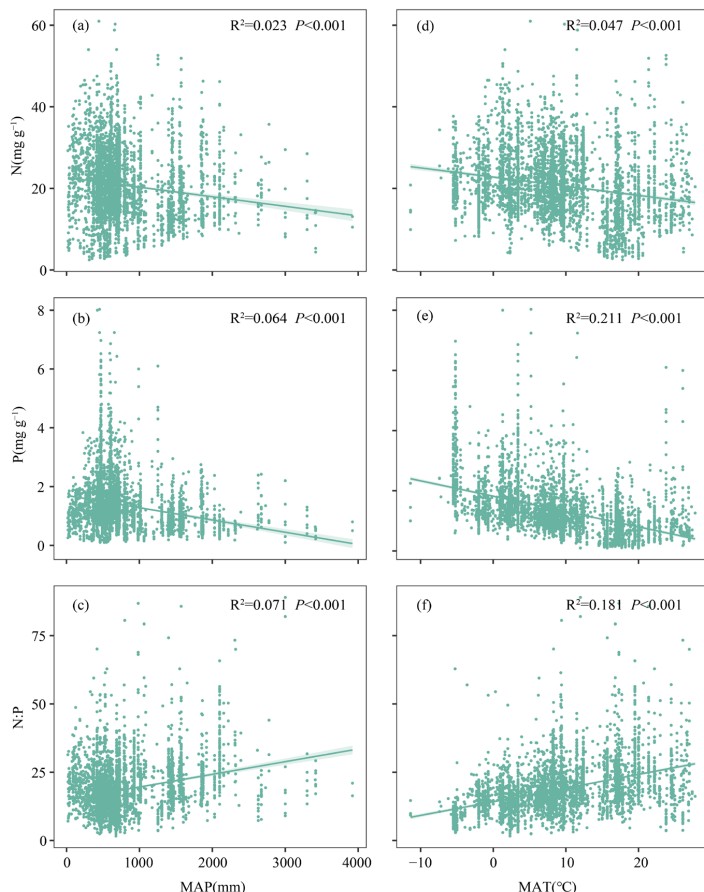


**Figure 4:** Relationship between leaf N, P and N:P ratio and annual mean precipitation and annual mean temperature of shrub leaves. The
data are sourced from Tian et al. (2019), Li et al. (2021), Qin et al. (2022), Dong et al. (2023a), and Dong et al. (2023b).
**4.3 Phylogenetic Relationships, Functional Types, and Age Effects**
Apart from environmental factors, an increasing number of studies have revealed a strong correlation between the variation in





stoichiometry and phylogenetic relatedness among plant species (He et al., 2006; Kerkhoff et al., 2006), supporting the
biogeochemical ecological niche hypothesis. This hypothesis posits that organisms require specific quantities and proportions
of essential nutrients to sustain growth. Due to differences in functional traits and life strategies, different species exhibit
distinct nutrient requirements and thus occupy varying positions and sizes within the n-dimensional space of multiple elemental
content (Peñuelas et al., 2008, 2019). For example, Sardans et al. (2021) analyzed the leaf concentration of N, P, and other
elements in 2,3962 trees from 227 species and found that shared ancestry explained 60-94% of the total variation in leaf
nutrient concentration and ratios, while current climate, atmospheric nitrogen deposition, and soil types collectively explained
1-7%. He et al. (2006) studied 213 species in Chinese grasslands and found that genus-level congruence explained 58.77% of
total variation in leaf N and C:N, while growing-season mean temperature and precipitation explained less than 3%. Yang et
al. (2017) examined *Artemisia* species in northern China and demonstrated that species identity accounted for over 30% of the
total variance in C, N, P, and their stoichiometric ratios. Vallicrosa et al. (2022) discovered that the variation in leaf N and P
content of forests globally was primarily driven by evolutionary history rather than by environmental factors. However, some
studies have reported inconsistent results. For instance, He et al. (2010) reported that phylogeny and environment explained
roughly equal proportions of the variation in N content in Chinese grasslands. Zhang et al. (2012) investigated ten elements in
the leaves of 702 terrestrial plant species in China and found that environmental factors outweighed phylogeny in explaining
the variation in leaf N and P content. An et al. (2021), analyzing leaf traits of terrestrial plants in China, argued that
environmental factors explained a larger proportion of the variation in N and P content (44.4% to 65.5%) compared to
phylogeny (3.9% to 23.3%). Conversely, Akram et al. (2023) found that the leaf C, N, and P content of dominant plant species
were relatively stable in arid deserts in China, with minimal influence from both, environment and phylogeny (explanatory
rates below 15%). Therefore, the influence of phylogenetic factors on leaf stoichiometry remains inconclusive. Although few
studies have examined the impact of shrubs phylogeny on their ecological stoichiometry, they all agree that phylogeny plays
a crucial role in regulating the variation of shrub stoichiometry, particularly for N. For example, Li et al. (2021) observed a
significant phylogenetic signal in leaf N content but not in C or P content among shrubs in the southwestern karst region.
Akram et al. (2020) found significant phylogenetic signals in both leaf C and N content of desert shrubs in northwest Gansu
Province. Yang et al. (2016), studying 163 shrub species in northern China, reported a strong phylogenetic signal in leaf N
content but a weaker signal in P. Liu et al. (2013) pointed out that phylogeny was the primary factor driving the variation in
leaf N and P content of shrubs in central Inner Mongolia, explaining 48% and 29.6% of the variation in N and P content,
respectively. These data suggest that the influence of species phylogeny should be thoroughly considered in future studies on
shrub stoichiometry.

Different shrub functional types also impact leaf C:N:P ratios. Numerous studies have shown that differences in shrub

functional types exert a greater impact on leaf stoichiometry than factors such as climate and soil (Niu et al., 2013; Ning et al.,



2019; Luo et al., 2017; Wang et al., 2020; Zhao et al., 2018; Zhang et al., 2018; Zou et al., 2021; Zhang et al., 2022). Generally,
evergreen shrubs tend to have higher C, C:N, and C:P ratios along with lower N, P, and N:P ratios in their leaves than deciduous
shrubs (Duan, 2023; Guo et al., 2021; Jing et al., 2017; Pi et al., 2017; Wang et al., 2022c; Zhao et al., 2018; Zhang et al., 2018;
Zhang et al., 2022). Moreover, there are reports of stronger correlations between the leaf C:N:P ratios of deciduous shrubs and
environmental factors (Wang et al., 2022c), suggesting that deciduous shrubs may be more sensitive to environmental changes
(Zhang et al., 2018). Regarding nitrogen-fixing species (such as leguminous shrubs), N, P, and N:P are higher in leguminous
shrubs than in non-leguminous shrubs (Akram et al., 2020; Guo et al., 2017; Ning et al., 2019; Tao et al., 2016; Zhang et al.,
2018), attributed to their N fixation capability (Vitousek et al., 2002). Leguminous shrubs also exhibit more stable N and P
stoichiometry than non-leguminous shrubs (Guo et al., 2017; Zhang et al., 2018).
Mycorrhizal type also impact the stoichiometry of shrubs. Chen et al. (2021) studied shrublands in peatlands in Northeast
China and found that shrubs with ericoid mycorrhizae had higher C, C:N, C:P, and N:P ratios in their leaves, but lower N and
P content, than those with ectomycorrhiza. Yang et al. (2021) analyzed the C, N, and P stoichiometry of shrubs from 725 plots
in northern China under different mycorrhizal types and reported that shrubs with ectomycorrhiza reflected higher P content
in their leaves than those with arbuscular mycorrhizae, while N content did not differ significantly. This can be attributed to
the fact that ectomycorrhizae have stronger phosphorus absorption than arbuscular mycorrhizae (Zhang et al., 2018; Toussaint
et al., 2020).
Individual differences arising from different shrub ages can also influence leaf stoichiometry, yet there is no consensus
on the direction of changes in leaf nutrient content with age. For instance, Zeng et al. (2017) studied the C, N, and P
stoichiometry in leaves of *Caragana korshinskii* on the Loess Plateau and found that leaf C, N, and P content increased with
shrub age, while C:N, C:P, and N:P ratios decreased. In contrast, Zhang et al. (2016) investigated the desert shrub *Haloxylon*
*ammodendron* in North China and observed that leaf C and N content, as well as the N:P ratio, rapidly increased with stand
age, while C:N significantly decreased, and P content and C:P ratio did not differ among age classes. Conversely, Dong et al.
(2023) studied the evergreen shrub *Ammopiptanthus mongolicus* in the western Ordos Plateau and reported that with increasing
shrub age, leaf N and P content decreased, while C:N, C:P, and N:P ratios increased, and C content remained relatively
unchanged.
**4.4 Impacts of Human Activities**
N and P elements are commonly recognized as limiting factors in terrestrial ecosystems (Elser et al., 2007), and the ecological
effects of N deposition have emerged as a pivotal research topic in global change ecology in recent years. Consequently,
fertilization experiments are frequently used by researchers to explore the impact of N deposition on nutrient cycling.
Experimental results indicated that nitrogen fertilization generally led to an increase in N content in shrub leaves, particularly





in temperate shrublands (Wang et al., 2017; Xu et al., 2021; Yu et al., 2023; Zhang et al., 2022). However, due to local variations
and experimental treatments, contrasting findings have also been reported. Some studies found that nitrogen application
enhances the N:P ratio in shrub leaves, further intensifying the phosphorus limitation (Kruk and Podbielska, 2018; Wang et al.,
2014; Zhang et al., 2022). A 14-year N addition experiment conducted by Vesala et al. (2021) in Scottish shrublands revealed
that N addition reduced fungal N supply to ericaceous mycorrhizal shrubs, thereby promoting P uptake by roots. In contrast,
Wang et al. (2016) observed no significant response of leaf N, P, or N:P in shrubs to N addition after a 5-year experiment in
subtropical evergreen broad-leaved forests in Wuyi Mountains, Fujian Province of China. When comparing shrubs of different
functional types, Yuan et al. (2021) found that the deciduous shrub *Vaccinium uliginosum* exhibited enhanced N and P uptake
capabilities under high nitrogen in the tundra belt of Changbai Mountains, while the evergreen shrub *Rhododendron aureum*
showed non-significant changes in leaf N and P, tending to maintain low plasticity. This may be attributed to the higher
homeostasis of evergreen shrubs (Sistla et al., 2015), resulting in a less sensitive response to fertilization and higher resilience
in nutrient-poor environments.

Disturbances from human activities such as logging and grazing can also impact the leaf C, N, and P of shrubs. Luiro et

al. (2010) found that logging significantly reduced N and P in shrub leaves, whereas Qiu et al. (2020) reported that logging
increased N and P content in shrub leaves under *Larix principis-rupprechtii* forests in North China, accompanied by decreases
in C:N and C:P ratios. In contrast, Pang et al. (2021) reported that logging did not alter C, N, P, or their stoichiometric ratios
in shrub leaves of secondary forests in the Qinling Mountains. These discrepancies may be related to the adaptive strategies of
different shrub species and the plant community recovery cycle following logging. Previous studies have shown that grazing
can increase N and P in shrub leaves. For instance, Wang et al. (2022) found that grazing enhanced N and P content in shrub
leaves in the meadow steppe of Hulunbeier, with C content peaking under moderate grazing intensity. García-Moreno et al.
(2014) observed that grazing led to increased N and P content in *Quercus ilex* leaves in the Mediterranean region, a finding
echoed by Baron et al. (2002). This may be attributed to the fact that light to moderate grazing stimulates plant growth by
removing shrub foliage to some extent, enhancing N and P content in leaves to promote photosynthesis and, consequently, aid
in resisting stressful environments (Han et al., 2008).
**5 Differences in Stoichiometry among Shrublands (Shrubs), Forests (Trees), and Grasslands (Herbs)**
The global geometric means of C, N, and P content, as well as C:N and N:P ratios, in terrestrial plant leaves are 464.0 mg g$^{-1}$,
18.93 mg g$^{-1}$, 1.20 mg g$^{-1}$, 22.5, and 15.8, respectively. Among these, shrub leaves have mean values of 454.66 mg g$^{-1}$, 18.86
mg g$^{-1}$, 1.18 mg g$^{-1}$, 23.4, and 16.1, while trees show 502.31 mg g$^{-1}$, 16.58 mg g$^{-1}$, 1.08 mg g$^{-1}$, 30.1, and 15.4, and herbaceous
plants 414.22 mg g$^{-1}$, 21.72 mg g$^{-1}$, 1.64 mg g$^{-1}$, 17.9, and 13.3 (Elser et al., 2000; Tian et al., 2018). In general, the N and P
content in leaves follow the order of herbs > shrubs > trees, whereas the N:P ratio follows the reverse order. This is attributed



to the accumulation of C in leaves of long-lived, slow-growing woody species to support their long growth and maintenance,
whereas short-lived, fast-growing herbaceous species require more N for growth and more P for their high proportion of
reproductive allocation (Aerts, 1996; Güsewell, 2004).
Upon conducting a comprehensive analysis of relevant research findings across different ecosystem types, it was observed
that the overall C content follows the trend of forest (483.42±31.35 mg g$^{-1}$) > shrubland (469.80±29.81 mg g$^{-1}$) > grassland
(438.00±30.20 mg g$^{-1}$) (He et al., 2006; Pang et al., 2020; Zhao et al., 2018). In contrast, the N content exhibits a pattern of
grassland (25.45±4.63 mg g$^{-1}$) > shrubland (20.42±3.10 mg g$^{-1}$) > forest (14.99±4.18 mg g$^{-1}$) (Fig. 5a), which is similar to the
global comparison among different growth forms. However, the differences in P content among these ecosystem types were
not significant (Fig. 5b). Notably, a few studies deviate from these general trends. For example, shrub leaves in southern China
reflect an N content of 16.4 mg g$^{-1}$ (Zhang et al., 2018), which is lower than that of most forest ecosystems, while forests in
eastern China show a relatively high N content (24.49 mg g$^{-1}$, Zhao et al., 2016), comparable to many grassland ecosystems.
Given the pronounced differences in N but not in P among ecosystems, the variation in N:P ratios across ecosystems is
primarily determined by N. Specifically, the N:P ratio generally follows the order of shrubland (17.93±7.53) ≈ grassland
(17.28±4.15) > forest (10.18±3.37) (Fig. 5c), suggesting that shrubland and grassland ecosystems may be more constrained by
P, whereas forest ecosystems are primarily limited by N.

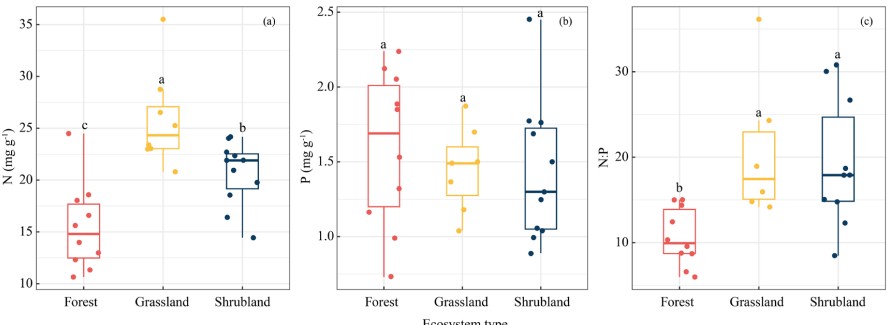


**Figure 5:** Differences in leaf (a) N, (b) P, and (c) N:P stoichiometry in forest, grassland and shrubland. [**] indicates statistical significance at
$P < 0.01$; [***] indicates statistical significance at $P < 0.001$. This figure is replotted using data from He et al. (2010), Wu et al. (2012), Liu et
al. (2013), Yang et al. (2016), Zhao et al. (2016), Zhang et al. (2017), Wang et al. (2018), Zhang et al. (2018), Zhao et al. (2018), Sun et al.
(2019), Pang et al. (2020), Liu et al. (2021), Rawat et al. (2021), Shi et al. (2021), Xu et al. (2021), Zhang et al. (2021), Chen et al. (2022),
Qin et al. (2022), Wang et al. (2022d), Bagedeng et al. (2023), Dong et al. (2023) and Lu et al. (2023).
When focusing on shrubland ecosystems, the N content in shrub leaves is generally higher than that in herbaceous species
(Fig. 6a). This can be attributed to the fact that shrubs, particularly in arid regions, are mostly deciduous. On the one hand,
deciduous shrubs have shorter leaf lifespans than evergreen shrubs, typically exhibiting higher N content and growth rates
(Zhang et al., 2014). On the other hand, leguminous shrubs can fix atmospheric $N_2$ through symbiotic rhizobia (Vitousek et al.,
2002), resulting in higher N content. As depicted in Fig. 6a, Dong et al. (2023) found that the N content in leaves of the



leguminous shrub *Sophora moorcroftiana* in the Yarlung Zangbo river was 21.78% higher than that of herbaceous plants. Jing
et al. (2017) discovered that the N content in the leaves of *Indigofera tinctoria* and *Lespedeza bicolor*, both leguminous shrubs
in the rocky dry area of southwestern Hunan, was 30.19% higher than that in herbaceous species. Chen et al. (2022) studied
*Caragana brevifolia* and the deciduous shrub *Dasiphora fruticosa* in eastern Qinghai-Tibet Plateau and found that the N
content in shrub leaves was 38.56% higher than that in the underlying herbaceous vegetation. By contrast, the P content does
not exhibit a consistent pattern among different life forms (Fig. 6b). Some studies have found higher P content in herbaceous
plants (Akram et al., 2020; Dong et al., 2023; Guo et al., 2022; Zhang et al., 2014), while others have reported higher P content
in woody species (Bagedeng et al., 2023; Dong et al., 2021; Umair et al., 2020; Zhang et al., 2023). This disparity can be
explained by the fact that P in leaves exists not only as nucleic acids but also in inorganic forms (orthophosphates), which
often reflect soil P availability (Oyarzabal and Oesterheld, 2009; Sterner and Elser, 2002). Therefore, leaf P content is
influenced by both plant life forms and environmental factors such as soil available P. Due to the typically higher N content in
shrubs, the N:P ratio tends to be lower in herbaceous plants than in shrubs (Fig. 5c), suggesting that herbaceous plants in
shrubland ecosystems are relatively more N-limited.

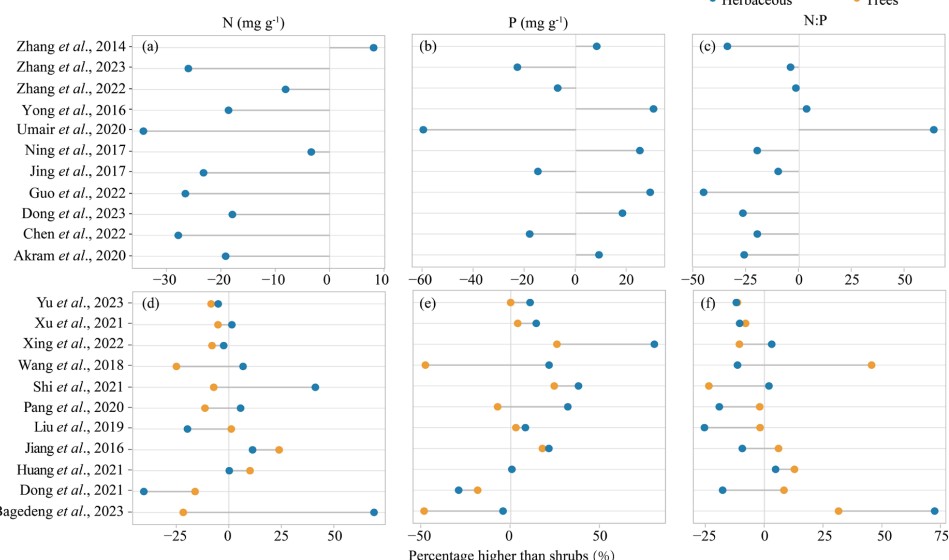


**Figure 6:** Differences in leaf N: P stoichiometry of shrubs, trees and herbaceous. (a, b, c) Shrubland ecosystem. (d, e, f) Forest ecosystem.
**6 Conclusion and Future Prospects**
With the increasing recognition of the importance of shrublands and shrubs for ecosystem functioning, research into their
stoichiometry has also increased, providing a more comprehensive understanding of the response and adaptive mechanisms of
different vegetation types to climate change. Current research primarily focused on the impacts of soil properties, climatic
factors, phylogenetic and individual differences, as well as human activities on the C, N, and P stoichiometry of shrub leaves,



yielding numerous insights. However, compared to studies on forest and grassland ecosystems, research in this area remains
relatively scant. It is recommended to strengthen research in the following aspects to gain a deeper insight into shrublands, as
unique ecosystems.

### 6.1 Strengthen research on stems, reproductive organs, and other plant organs

The results of our literature analysis showed that current research on the stoichiometry of shrublands -and shrubs- primarily
focused on leaves, with a limited number of reports also examining roots and litter. However, there is a significant lack of
knowledge regarding the stoichiometry of stems, branches, and reproductive organs, as well as their environmental control
mechanisms. Consequently, there is an even more pronounced deficiency in studies exploring the inter-organ correlation of
elements. Addressing these gaps would enhance our understanding of shrubs' allocation patterns and trade-off strategies for
different elements, thereby providing deeper insights into the ecological adaptation of shrubs and shrublands in response to
potential environmental changes.

### 6.2 Thoroughly consider the influence of multiple factors on shrub stoichiometry

Stoichiometry is species-specific, with plant element content continuously balancing between phylogenetic constraints and
environmental influences to adapt to changing environments (Qu et al., 2024). Nevertheless, the extent to which species
differences, i.e., phylogenetic levels, influence stoichiometry of shrubs remains understudied, needing further research. In
terms of environmental impacts, studies have primarily focused on temperature, precipitation, and soil factors, yet research on
the effects of atmospheric components such as carbon dioxide changes, UV radiation, N deposition, and saline-alkali stress on
stoichiometry of shrubs and shrublands is scarce. Currently, the relative influence of environmental factors and phylogeny on
leaf stoichiometry remains unclear, and the choice of different methodological approaches can lead to starkly contrasting
interpretations (Tian et al., 2024). By selecting appropriate data analysis models and integrating considerations of multiple
environmental impacts, we can gain a more comprehensive understanding of the adaptation and evolution mechanisms of
shrub species under extreme habitat conditions such as alpine and arid environments.

### 6.3 Conduct coupled research on the aboveground-belowground stoichiometry of shrub ecosystems

The coupling relationship among soil microorganisms, soil properties, and plant stoichiometry represents one of the hotspots
and challenges in ecological research. Shrublands often occur in habitats with relatively extreme water and/or temperature
conditions, where biogeochemical cycling processes exhibit stronger dependence on microorganisms. However, there is a
significant lack of research on the coupling relationship between belowground processes and plant stoichiometry in shrublands.
This limitation has greatly constrained our understanding of how shrub species adapt to stressful habitats like alpine and arid



environments.

**6.4 Strengthen research on the evolution of stoichiometry during shrub encroachment processes**

Literature data reveals that shrublands and shrubs show distinct stoichiometry compared to forests (trees) and grasslands (herbaceous plants), generally showing intermediate levels of N and P content, and their ratios, between forests and grasslands. Given the widespread occurrence of shrub encroachment globally, the evolution of vegetation types or shifts in dominance among species with different life forms inherently entails changes in community stoichiometry. Conducting long-term, continuous monitoring studies on the stoichiometric dynamics of shrub ecosystems will facilitate a deeper understanding of the biogeochemical cycling mechanisms of shrub encroachment processes, thereby providing mechanistic explanations for shrub encroachment phenomena and processes in the context of global change.

**CRediT authorship contribution statement**

The study was conceptualized by Lin Zhang. Xin-Ru Zhang wrote the first draft of the paper. Zhong Wang, Jin-Niu Wang and Lin Zhang revised the manuscript. All authors have read and approved the published version of the manuscript.

**Data availability**

The raw data and R code that supporting the findings of this study are available on figshare at: https://figshare.com/s/9889881b33d3565e5a19.

**Declaration of Competing Interest**

The authors declare that they have no known competing financial interests or personal relationships that could have appeared to influence the work reported in this paper.

**Acknowledgments**

This work was supported by Natural Science Foundation of Tibet Autonomous Region (XZ202401JD0025).

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
