# Peer review of "and Phosphorus in Shrubs and Shrublands"

_EGUsphere, 2025_

## Author Comment (AC1)

**[Comment 1]**: The abstract could be refined to highlight more clearly the quantitative outcomes of the synthesis (e.g., mean global leaf C, N, P values and N:P ratios across life forms).

**[Response]**: Thanks for the suggestions. In this version, we have revised the abstract accordingly as follows: "**Abstract:** Ecological stoichiometry examines the balance and ratios of multiple elements in ecological processes. In shrubs, characterized by their adaptability to extreme environments such as alpine and arid regions, stoichiometric traits likely differ from those in trees and grasses, reflecting unique ecological strategies. However, this hypothesis remains insufficiently explored. Here we review the current state of shrub stoichiometry and then identify research hotspots of shrub stoichiometry. Then, we summarize the effects of climate, soil properties, phylogeny, ontogeny, and human activities on stoichiometry of shrub leaves. In addition, we compared the stoichiometry of shrublands with that of forests and grasslands. The development of shrub stoichiometry research can be broadly divided into three stages: the initial stage, the rapid development stage, and the high-quality development stage. Notably, existing studies have primarily focused on leaf stoichiometry. Mean values of C, N, P, C:N, and N:P in shrub leaves globally were 454.66 mg g⁻¹, 18.93 mg g⁻¹, 1.20 mg g⁻¹, 23.4, and 16.1, respectively. Shrub leaf N and P content were higher than those of trees (16.58 mg g⁻¹, 1.18 mg g⁻¹, respectively) and lower than herbs (21.72 mg g⁻¹, 1.64 mg g⁻¹, respectively). In contrast, C content and C:N ratio showed opposite trends, being lower than trees (502.31 mg g⁻¹, 30.1) but higher than herbs (414.22 mg g⁻¹, 17.9). Importantly, the N:P ratio in shrub leaves exceeded that of both trees (15.4) and herbs (13.3), suggesting stronger P limitation in shrubs. Leaf N and P content correlated positively with soil nutrients and precipitation, and negatively with temperature. Functional types also influenced stoichiometry, with deciduous and leguminous shrub species showing higher N and P content than evergreen and non-leguminous shrubs. Future studies should integrate above- and below-ground stoichiometry, consider phylogenetic influences, and standardized sampling and analytical protocols to better understand shrubland adaptation and

formation mechanisms under global change."

: The section discussing phylogenetic influences is quite dense; a table summarizing key findings from major studies could improve accessibility.

**[Response]:** We have revised this section and added Table 1 accordingly. The revised content is presented as follows.

"**4.3 Phylogenetic Relationships, Functional Types, and Age Effects**

Apart from environmental factors, an increasing number of studies have revealed a strong correlation between the variation in stoichiometry and phylogenetic relatedness among plant species (He et al., 2006; Kerkhoff et al., 2006), supporting the biogeochemical ecological niche hypothesis. This hypothesis posits that organisms require specific quantities and proportions of essential nutrients to sustain growth. Due to differences in functional traits and life strategies, different species exhibit distinct nutrient requirements and thus occupy varying positions and sizes within the n-dimensional space of multiple elemental content (Peñuelas et al., 2008, 2019). For example, Sardans et al. (2021) analyzed the leaf concentration of N, P, and other elements in 2,3962 trees from 227 species and found that shared ancestry explained 60-94% of the total variation in leaf nutrient concentration and ratios, while current climate, atmospheric nitrogen deposition, and soil types collectively explained 1-7%. Similar findings have been reported at regional scales; however, some studies have yielded contrasting results (Table 1). Therefore, the influence of phylogenetic factors on leaf stoichiometry remains inconclusive. Although few studies have examined the impact of shrubs phylogeny on their ecological stoichiometry, they all agree that phylogeny plays a crucial role in regulating the variation of shrub stoichiometry (Table 1), particularly for N content. Studies on shrub elemental stoichiometry in the southwestern karst region of China (Li et al., 2021;), Gansu Province (Akram et al., 2020), and northern China (Yang et al., 2016) have consistently reported strong phylogenetic signals in leaf N. These findings suggest that species phylogeny should

be carefully considered in future research on shrub stoichiometry, particularly when a large number of species are involved."

Table 1 Summary of studies on phylogenetic and environmental effects on leaf elements.

| Study area | Vegetation type | Species (n) | Elements | Variance explained by phylogeny | Variance explained by environment | Reference |
|---|---|---|---|---|---|---|
| Global | Forests | 227 | N, P, K, Ca, Mg and S | 60–94% | 1–7% | Sardans et al. 2021 |
| Global | Forests | 2,000 | N, P, K | >60% | – | Vallicrosa et al. 2022 |
| China | Grasslands | 213 | N, C:N | 58.8% | <3% | He et al. 2006 |
| China | Grasslands | 147 | N | 36% | 38% | He et al. 2010 |
| China | Woody plants | 702 | N, P | 16–38% | 42–55% | Zhang et al. 2012 |
| China | Woody plants | 3,000 | N, P | 3.9–23.3% | 44.4–65.5% | An et al. 2021 |
| Arid deserts, China | Woody plants | 15 | C:N:P stoichiometry | 1.8–54.2% | 3.6–66.3% | Akram et al. 2023 |
| Inner Mongolia, China | Shrubs | 55 | N,P and N:P ratio | 29.6–48% | <11% | Liu et al. 2013 |
| China | Shrubs | 11 | C:N:P stoichiometry | 32.9–40.3% | 17.0–19.0% | Yang et al. 2015 |

[Comment 3]: Consider expanding on methodological challenges (e.g., variation in sampling organs, geographic biases, or digitization uncertainty) that might affect comparability across studies.

[Response]: Thanks for this valuable suggestion. We have integrated this content into the Conclusion and Future Prospects, as shown below.

 "**6.5 Methodological Considerations for Future Research**

Future studies should place greater emphasis on ensuring methodological consistency. Differences in sampling organs, geographic sampling biases, statistical approaches, and uncertainties related to data extraction from published sources can all reduce the comparability of results across studies and affect their accuracy. For example, root elemental concentrations can vary substantially between fine roots, coarse roots, and roots of different diameter classes, making it difficult to compare findings across studies without consistent sampling protocols (You et al., 2023). Similarly, using different analytical methods—such as random forest models versus linear mixed-effects models—may lead to opposite conclusions about the relative importance of

phylogeny and environment in shaping leaf stoichiometry, depending on the data structure and model assumptions (Tian et al., 2024). Therefore, developing standardized protocols for sample collection, spatial representation, and data reporting is essential to improve the reliability, comparability, and overall synthesis of shrub stoichiometry research across regions and species."

---

## Author Comment (AC2)

**Specific comments:**

**[Comment 1]**: Abstract: The outlook for the future research directions of shrub chemometrics in the abstract section is relatively simple. This part should be further refined in combination with the conclusion section to make it more specific.

**[Response]**: Thank you for the helpful suggestions. We also incorporated Reviewer 1's suggestion ("The abstract could be refined to highlight more clearly the quantitative outcomes of the synthesis"), and have revised the abstract accordingly as follows: "**Abstract:** Ecological stoichiometry examines the balance and ratios of multiple elements in ecological processes. In shrubs, characterized by their adaptability to extreme environments such as alpine and arid regions, stoichiometric traits likely differ from those in trees and grasses, reflecting unique ecological strategies. However, this hypothesis remains insufficiently explored. Here we review the current state of shrub stoichiometry and then identify research hotspots of shrub stoichiometry. Then, we summarize the effects of climate, soil properties, phylogeny, ontogeny, and human activities on stoichiometry of shrub leaves. In addition, we compared the stoichiometry of shrublands with that of forests and grasslands. The development of shrub stoichiometry research can be broadly divided into three stages: the initial stage, the rapid development stage, and the high-quality development stage. Notably, existing studies have primarily focused on leaf stoichiometry. Mean values of C, N, P, C:N, and N:P in shrub leaves globally were 454.66 mg g$^{-1}$, 18.93 mg g$^{-1}$, 1.20 mg g$^{-1}$, 23.40, and 16.10, respectively. Shrub leaf N and P content were higher than those of trees (16.58 mg g$^{-1}$, 1.18 mg g$^{-1}$, respectively) and lower than herbs (21.72 mg g$^{-1}$, 1.64 mg g$^{-1}$, respectively). In contrast, C content and C:N ratio showed opposite trends, being lower than trees (502.31 mg g$^{-1}$, 30.10) but higher than herbs (414.22 mg g$^{-1}$, 17.90). Importantly, the N:P ratio in shrub leaves exceeded that of both trees (15.40) and herbs (13.30), suggesting stronger P limitation in shrubs. Leaf N and P content correlated positively with soil nutrients and precipitation, and negatively with temperature. Functional types also influenced stoichiometry, with

deciduous and leguminous shrub species showing higher N and P content than evergreen and non-leguminous shrubs. Future research should integrate nutrient dynamics and stoichiometric traits of both above- and below-ground organs. Increased attention should be given to phylogenetic constraints on nutrient allocation, and shrub stoichiometry should be incorporated into predictive models of ecosystem nutrient cycling. Moreover, the standardization of methodologies in sampling, organ selection, and data analysis is essential to enhance cross-study comparability. These efforts will facilitate a deeper understanding of the adaptive strategies and formation mechanisms of shrubland ecosystems under global environmental change."

[Comment 2]: "2.1 Literature and Data Collection": It is suggested that the retrieval flowchart of the review literature database be supplemented in this section to visually clarify the steps of literature retrieval and screening. This can further confirm the accuracy and validity of the literature retrieval results.

[Response]: We have added a flow diagram detailing the literature search and screening procedures, as shown in Figure 2:

[Figure]

**Figure 2:** Flow diagram of the literature screening process.

[**Comment 3**]: "3 The Historical Development and Research Hotspots of Chemical Stoichiometry in Shrubs": This part is a bit complicated. It is suggested to add a subtitle (3.1,3.2...) to organize and elaborate on the content of this section. The elaboration on the research hotspots merely conducts a statistical classification of the review literature and lists some references, lacking in-depth analysis and summary of the research hotspots. It is suggested that this part be further improved.

[**Response**]: Thank you for the valuable comments on Section 3 ("The Historical Development and Research Hotspots of Chemical Stoichiometry in Shrubs"). Incorporating Comments 3 and 4, we have further reorganized and refined; the revised text is as follows:

"**3 The Historical Development and Research Hotspots of Chemical Stoichiometry in Shrubs**

**3.1 Historical Development of Shrub Stoichiometry Research**

The entire developmental history can be broadly divided into three stages: the initial development stage (before 2010), the fast development stage (2011-2018), and the high-quality development stage (2019-2023) (Fig. 3). Each stage has demonstrated clear changes in research focuses, theoretical mechanisms, and geographic emphasis.

During the initial stage (before 2010), shrub-related stoichiometric research was scarce, with only 67 publications recorded globally. The field was still in its infancy, and shrubs were often studied as part of forest (e.g., tall shrublands or understory shrubs) or grassland ecosystems (companion shrubs embedded in grassland ecosystems) rather than as independent functional types. The seminal work by Castro-Díez et al. (1997), which revealed precipitation-driven variation in leaf N and N:P ratios in *Quercus coccifera*, marked one of the earliest targeted explorations of shrub stoichiometry. The publication of "Ecological Stoichiometry: The Biology of Elements from Molecules to the Biosphere" by Sterner and Elser (2002) provided a theoretical foundation for future development, but the concept had yet to gain traction in China due to its relatively late introduction.

From 2011 to 2018, research on shrub stoichiometry entered a phase of rapid development. Internationally, the focus gradually shifted from descriptive studies of species-level nutrient traits to investigating their regulatory roles in ecosystem carbon and nutrient cycling. Sistla and Schimel (2012) proposed that stoichiometric flexibility could serve as a regulatory factor for changes in carbon and nutrient cycling in terrestrial ecosystems, marking a transition toward mechanism-oriented research. In China, the launch of national-level initiatives—such as Strategic Priority Research Program of the Chinese Academy of Sciences on "Current Status, Rates, Mechanisms, and Potential of Carbon Sequestration in Shrub Ecosystems" (2011), and the Ministry of Science and Technology of China initiated the Special Project for Basic Scientific Research entitled "Survey of Shrub Plant Communities in China"

(2015), significantly stimulated domestic research activities. Chinese scholars began to explore emerging questions, including seasonal dynamics and nutrient allocation among plant organs. For instance, Niu et al. (2013) analysed the seasonal variation patterns of C:N:P stoichiometry in leaves of major shrub species in the Alxa Desert, finding that leaf C and N content and C:N ratios varied little, while P, C:P, and N:P ratios showed greater variability. He et al. (2017) discovered differences in C, N, and P content among different organs of *Sibiraea angustata* in eastern Qinghai-Tibet Plateau, partially supporting the elemental homeostasis theory and the growth rate hypothesis.

Since 2019, research has entered a high-quality development phase characterized by thematic diversification and theoretical innovation. The volume of publications has consistently exceeded 50 per year, with Chinese-authored articles accounting for nearly half of the English-language output, indicating a geographical shift in leadership. Research emphasis has expanded to include the functional implications of stoichiometry at the community and ecosystem levels. The well-coordinated elements hypothesis illustrates this trend by highlighting the mutual constraint of functionally linked elements (Zuo et al., 2024). Studies have also integrated stoichiometry with biodiversity metrics and ecosystem transitions; for example, Song et al. (2021) associated shrub diversity with foliar N and P concentrations, while Urbina et al. (2020) examined the biogeochemical consequences of shrub encroachment. These advancements have been supported by large-scale national initiatives, such as the Second Tibetan Plateau Comprehensive Scientific Expedition Research Program launched by the Ministry of Science and Technology of the People's Republic of China in 2019, and the Xinjiang Comprehensive Scientific Expedition Project in 2022, both of which have significantly elevated the visibility and importance of stoichiometric research on shrubland ecosystems, particularly in desert environments.

[Figure]

**Figure 3:** Changes in the number of papers published in the field of ecological stoichiometric studies of shrublands and shrubs from 1997 to 2023. The pie chart depicts the research subjects and their respective proportions in the literature on shrub ecological stoichiometry.

**3.2 Main Research Subjects of Shrub Stoichiometry**

Among the 540 selected research articles in Chinese, 34.6% focused on the C, N, P stoichiometry of a specific organ in shrubs; most (80.75%) focused on leaves, a smaller proportion (9.10%) on roots, and yet another minority (8.56%) focused on litter, while only a few papers encompassed branches, bark, and seeds. Out of the 540 articles, 28.50% targeted shrubland soil, 25.90% addressed both shrubs and soil, and merely a 10.90% examined the stoichiometry across multiple shrub organs (Fig.2 inset).

Prior research on shrub organs primarily centered on geographical patterns of stoichiometric traits. Zhao et al. (2018) revealed that alpine shrub leaves in river valleys had higher N and P contentand lower C content, with the latter increasing with elevation and decreasing temperature. You et al. (2023) reported that shrub root C and N content increased with latitude, while N content decreased with age/root diameter. Studies examining both shrubs and soil emphasized their reciprocal influence on

stoichiometry. For instance, Müller et al. (2017) observed significant positive correlations between C:N, C:P, and N:P ratios in *Rhododendron campanulatum* and soil stoichiometry. Additionally, vegetation type has been found to impact soil stoichiometry, with higher N and P content in soil under *Caragana korshinskii* than under *Hippophae rhamnoides*, and soil C, N, P content and their stoichiometric ratios correlated with density of litter per square meter and root mass/volume (Wang et al., 2022a). Research on multiple shrub organs provided deeper insights into shrub nutrient allocation strategies and environmental adaptability. For example, Dong et al. (2023a) found that N and P content in shrubs generally followed the order of seeds > flowers > leaves > roots > stems, indicating that shrubs allocate more nutrients to reproductive organs like seeds and flowers to enhance reproduction efficiency. Yang et al. (2014) explored N and P allocation strategies in leaves, stems, and roots of shrubs in northern China, revealing that N content ratios among organs exhibited allometric growth between leaves and non-leaf organs and isometric growth among non-leaf organs, while P concentrations tended towards allometric growth between roots and non-root organs and isometric growth among non-root organs.

**3.3 Research Hotspots and Trends**

Through the analysis of the keyword clustering map (Fig. 3), we found that research on the ecological stoichiometry of shrubs primarily focuses on eight aspects: ecological stoichiometry, foliar nutrients, drought stress, leaf morphology, facilitation, allometric growth, community structure, and climate change. Keywords with higher frequencies include nitrogen, carbon, growth and soil, indicating that the research mainly focused on nutrient limitations in shrub growth and the relationship between the stoichiometry of shrubs and soil (Fig. 4). A closer analysis of the keywords and associated literature reveals a set of evolving research priorities and emerging thematic trends:

**3.3.1 Nutrient limitation mechanisms and their environmental drivers**

Research is gradually shifting from static descriptions of C, N, and P ratios toward

exploring the underlying mechanisms that constrain shrub growth. For example, He et al. (2023) discovered that the leaf C content of alpine shrub plants was higher than the global average for plants, and their growth was primarily constrained by N, suggesting that cold climates influence nutrient acquisition strategies in shrubs. In recent years, an increasing number of studies have incorporated elevation, temperature, precipitation, and soil properties as explanatory variables (Liu et al., 2023; Lu et al., 2023), aiming to investigate shrub responses to environmental changes from the perspective of nutrient utilization.

**3.3.2 Leaf traits and stoichiometric trade-offs under stress conditions**

Foliar stoichiometry has been widely investigated under drought and temperature stresses, particularly in relation to other functional traits such as specific leaf area (SLA) and leaf thickness (Zhang et al., 2017; Jiang et al., 2021). These studies aim to elucidate how shrubs coordinate carbon acquisition and water conservation strategies in response to environmental constraints, thereby contributing to the development of trait-based stoichiometric frameworks.

**3.3.3 Shrub–soil feedbacks and biogeochemical cycling**

In recent years, increasing attention has been directed toward the bidirectional feedbacks between shrub stoichiometry and soil nutrient status. A growing body of research highlights that root traits (You et al., 2023), soil moisture (He et al., 2023), bulk density (Luo et al., 2022), litter decomposition, and microbial processes jointly drive the coupled stoichiometric dynamics of plants and soils. These studies collectively underscore the importance of viewing stoichiometric traits as integral components of nutrient cycling in terrestrial ecosystems.

**3.3.4 Intraspecific and interspecific variation across scales**

Recent research highlights substantial stoichiometric variation within and among shrub species, driven by both phylogenetic constraints and local adaptation. This has led to increasing attention to scaling stoichiometric patterns from organ to population, community, and biome levels (Zhao et al., 2016; Zhang et al., 2020).

In summary, the field is shifting toward a more mechanistic and integrative understanding of shrub stoichiometry, emphasizing multiscale interactions between plant traits, soil processes, and climatic factors. Future studies are expected to adopt more holistic approaches, including trait networks, experimental manipulations, and modeling, to deepen insights into shrub responses under global environmental change.

[Figure]

**Figure 4:** Keyword clustering map of ecological stoichiometry literature in shrublands and shrubs.

**3.4 Research Methods and Analytical Approaches**

Current research on shrub stoichiometry primarily relies on data obtained through field sampling and the integration of published literature. In field-based studies, the concentrations of C, N, and P in various plant organs (e.g., leaves, stems, roots) and soils are typically measured using elemental analyzers via dry combustion for C and N. P concentrations are commonly determined using the molybdenum blue colorimetric method or inductively coupled plasma mass spectrometry (ICP-MS).

For data analysis, a variety of multivariate statistical methods are employed to

explore stoichiometric patterns and their environmental or functional correlations. These include general linear regression, principal component analysis (PCA), structural equation modeling (SEM), and linear mixed-effects models (LMM). Such methods allow for disentangling the effects of climate, soil properties, and plant functional traits on stoichiometric variation, and are increasingly used to assess both direct and indirect drivers across spatial and ecological gradients."

[Comment 4]: "3 The Historical Development and Research Hotspots of Chemical Stoichiometry in Shrubs": The content of the research progress section should avoid simply listing literature. It is necessary to further sort out the development context of the research and enhance the logic of the evolution of this research field. In addition, this section lacks an overview of the research methods and is recommended to be supplemented.

[Response]: We have revised Section 3 ("The Historical Development and Research Hotspots of Chemical Stoichiometry in Shrubs"). Please see our response to Comment 3 for details.

[Comment 5]: "4.2 Climate Factors" and "4.3 Phylogenetic Relationships, Functional Types, and Age Effects": It is suggested to add three-level headings (4.2.1…,4.3.1…) to elaborate on the content point by point, making the article structure clearer.

[Response]: Thank you for the suggestion. We have added third-level subheadings (e.g., 4.2.1, 4.3.1) to Sections 4.2 and 4.3 and refined the logical flow to improve structural clarity. In addition, in line with Reviewer 1's suggestion, we have added Table 1. The revised content is as follows:

"**4.2 Climate Factors**

Plants take up essential nutrients from the soil, and climate plays a pivotal role in soil

development at larger scales, indirectly modulating nutrient cycling by influencing the formation, decomposition, and storage of soil organic matter (Mou et al., 2022; Ren et al., 2017), subsequently regulating ecological stoichiometry. Therefore, we focus primarily on how precipitation and temperature, the two dominant climatic factors, collectively shape the C:N:P stoichiometry patterns of shrub leaves at both global and regional scales.

**4.2.1 Precipitation effects on shrub leaf stoichiometry**

Precipitation exerts a major control on soil moisture, nutrient leaching, and biological productivity, thereby influencing plant nutrient concentrations. Globally, the N and P content in terrestrial plant leaves generally decrease with increasing annual precipitation (Reich and Oleksyn, 2004; Tang et al., 2018). We used a global dataset comprising leaf N and P of 4,253 shrubs content along with climatic factors, spanning 582 sites and 977 species (from Tian et al., 2019). By further integrating data from four recent publications, we found that leaf N and P content generally decreased with rising annual precipitation, while the N:P ratio increased significantly (Fig. 4a, 4b, 4c). Similar trends were observed at the regional scale, content such as the Loess Plateau (Zheng et al. 2007), or desert shrubs in Xinjiang Autonomous Region, China (He et al. 2019). Nevertheless, some studies have reported contrasting trends, with shrub leaf C, N, and P content increasing with precipitation (Dong et al., 2023b; Guo et al., 2021; Liu et al., 2013; Luo et al., 2022; Wang and Yu, 2017; Wang et al., 2019; Wang et al., 2022b, 2022c; Yang et al., 2019; Zhao et al., 2018). Discrepancies also exist in controlled experiments; for example, Prieto and Querejeta (2020) recorded a significant reduction in leaf N and P content after five years of water reduction in a Mediterranean semiarid shrubland, whereas Umair et al. (2020) found no changes in leaf N and P content with increasing water availability in a degraded karst system in southwestern China. These contrasting patterns of shrub leaf C, N, and P stoichiometry in response to precipitation may reflect their adaptive strategies to the environment. On one hand, high N content is considered an adaptation to arid regions (Wright et al., 2005), as it facilitates increased photosynthetic rates (Wright et al.,

2003). The negative correlation between leaf P content and precipitation is primarily attributed to soil P leaching under high moisture conditions (Chen et al., 2013; Ordoñez et al., 2009). On the other hand, shrubs are prevalent in arid and semiarid regions, where precipitation increases alleviate water limitation, prompting a shift towards investment-oriented growth, including elevated N to enhance photosynthesis (Liu et al., 2017; Wei et al., 2011) and increased P to accelerate growth rates (Luo et al., 2022).

**4.2.2 Temperature effects on shrub leaf stoichiometry**

Besides precipitation, previous studies have established that temperature exerted a significant influence on plant ecological stoichiometry, proposing two opposing hypotheses: the Temperature-Plant Physiology Hypothesis and the Temperature-Biogeochemistry Hypothesis (Reich and Oleksyn, 2004). The former suggests that temperature modulates plant physiological processes, leading to higher N and P content under low temperatures, whereas the latter posits that temperature influences plant N and P stoichiometry by altering soil N and P availability, resulting in lower leaf N and P content at lower temperatures (Reich and Oleksyn, 2004). On a global scale, trends in terrestrial plant leaf N and P content with temperature support the Temperature-Plant Physiology Hypothesis, indicating a decrease in leaf N and P content and an increase in N:P ratio with rising temperatures (Kang et al., 2010; Tang et al., 2018; Yuan and Chen, 2009). An analysis of published literature data reveals similar trends in global leaf N and P content of shrubs, with a significant negative correlation with mean annual temperature (Fig. 4c, 4d). At the regional level, most studies on shrub leaf stoichiometry have yielded similar results (He et al., 2019; Liu et al., 2013; Wang and Yu, 2017; Xu et al., 2021; Yang et al., 2016; Yang et al., 2019; Zhang et al., 2019). Short-term controlled experiments further show that warming reduces shrub leaf N (Prieto and Querejeta, 2020; Wu et al., 2019; Xu et al., 2024). However, some studies indicate that temperature may alter N and P mineralization rates by influencing soil microbial activity, resulting in lower shrub N and P content at lower temperatures, thereby establishing a positive correlation between temperature

and N, P content (Huang et al., 2021; Guo et al., 2021; Li et al., 2014; Wang et al., 2022b). Reich and Oleksyn (2004) reported a biphasic response of global plant leaf N and P content to temperature, initially increasing and then decreasing, with an inflection point around a mean annual temperature of 5°C. They suggested that the Temperature-Biogeochemistry Hypothesis dominates below 5°C, whereas the Temperature-Plant Physiology Hypothesis prevails above this threshold. For shrub leaf ecological stoichiometry, some studies support the Temperature-Biogeochemistry Hypothesis in regions with temperatures above 5°C (Huang et al., 2021; Guo et al., 2021; Li et al., 2014), but this has not been fully validated. Thus, while cases exist supporting the Temperature-Plant Physiology Hypothesis for the influence of temperature on shrub leaf stoichiometry, differences may arise due to variations in study species and regions, with the underlying reasons remaining unclear.

[Figure]

**Figure 5:** Relationship between leaf N, P and N:P ratio and annual mean precipitation and annual mean temperature of shrub leaves. The data are sourced from Tian et al. (2019), Li et al. (2021), Qin et al. (2022), Dong et al. (2023a), and Dong et al. (2023b).

**4.3 Biological Drivers**

In addition to environmental factors, increasing evidence suggests that plant ecological stoichiometric traits are closely associated with intrinsic biological characteristics, such as phylogenetic relatedness, functional types, mycorrhizal

associations, and age. The following section explores how these four factors influence the variation in leaf C:N:P stoichiometry.

**4.3.1 Phylogenetic influences**

Apart from environmental factors, an increasing number of studies have revealed a strong correlation between the variation in stoichiometry and phylogenetic relatedness among plant species (He et al., 2006; Kerkhoff et al., 2006), supporting the biogeochemical ecological niche hypothesis. This hypothesis posits that organisms require specific quantities and proportions of essential nutrients to sustain growth. Due to differences in functional traits and life strategies, different species exhibit distinct nutrient requirements and thus occupy varying positions and sizes within the n-dimensional space of multiple elemental content (Peñuelas et al., 2008, 2019). For example, Sardans et al. (2021) analyzed the leaf concentration of N, P, and other elements in 2,3962 trees from 227 species and found that shared ancestry explained 60-94% of the total variation in leaf nutrient concentration and ratios, while current climate, atmospheric nitrogen deposition, and soil types collectively explained 1-7%. Similar findings have been reported at regional scales; however, some studies have yielded contrasting results (Table 1). Therefore, the influence of phylogenetic factors on leaf stoichiometry remains inconclusive. Although few studies have examined the impact of shrubs phylogeny on their ecological stoichiometry, they all agree that phylogeny plays a crucial role in regulating the variation of shrub stoichiometry (Table 1), particularly for N content. Studies on shrub elemental stoichiometry in the southwestern karst region of China (Li et al., 2021), Gansu Province (Akram et al., 2020), and northern China (Yang et al., 2016) have consistently reported strong phylogenetic signals in leaf N. These findings suggest that species phylogeny should be carefully considered in future research on shrub stoichiometry, particularly when a large number of species are involved.

Table 1 Summary of studies on phylogenetic and environmental effects on leaf elements.

| Study area | Vegetation type | Species (n) | Elements | Variance explained by phylogeny | Variance explained by environment | Reference |
|---|---|---|---|---|---|---|
| Global | Forests | 227 | N, P, K, Ca, | 60–94% | 1.00–7.00% | Sardans et al. |

| | | | Mg and S | | | 2021 |
|---|---|---|---|---|---|---|
| Global | Forests | 2,000 | N, P, K | >60.00% | – | Vallicrosa et al. 2022 |
| China | Grasslands | 213 | N, C:N | 58.80% | <3.00% | He et al. 2006 |
| China | Grasslands | 147 | N | 36.00% | 38.00% | He et al. 2010 |
| China | Woody plants | 702 | N, P | 16.00–38.00% | 42.00–55.00% | Zhang et al. 2012 |
| China | Woody plants | 3,000 | N, P | 3.90–23.30% | 44.40–65.50% | An et al. 2021 |
| Arid deserts, China | Woody plants | 15 | C:N:P stoichiometry | 1.80–54.20% | 3.60–66.30% | Akram et al. 2023 |
| Inner Mongolia, China | Shrubs | 55 | N,P and N:P ratio | 29.60–48.00% | <11.00% | Liu et al. 2013 |
| China | Shrubs | 11 | C:N:P stoichiometry | 32.90–40.30% | 17.00–19.00% | Yang et al. 2015 |

**4.3.2 Functional types**

Different shrub functional types also impact leaf C:N:P ratios. Numerous studies have shown that differences in shrub functional types exert a greater impact on leaf stoichiometry than factors such as climate and soil (Niu et al., 2013; Ning et al., 2019; Luo et al., 2017; Wang et al., 2020; Zhao et al., 2018; Zhang et al., 2018; Zou et al., 2021; Zhang et al., 2022). Generally, evergreen shrubs tend to have higher C, C:N, and C:P ratios along with lower N, P, and N:P ratios in their leaves than deciduous shrubs (Duan, 2023; Guo et al., 2021; Jing et al., 2017; Pi et al., 2017; Wang et al., 2022c; Zhao et al., 2018; Zhang et al., 2018; Zhang et al., 2022). Moreover, there are reports of stronger correlations between the leaf C:N:P ratios of deciduous shrubs and environmental factors (Wang et al., 2022c), suggesting that deciduous shrubs may be more sensitive to environmental changes (Zhang et al., 2018). Regarding nitrogen-fixing species (such as leguminous shrubs), N, P, and N:P are higher in leguminous shrubs than in non-leguminous shrubs (Akram et al., 2020; Guo et al., 2017; Ning et al., 2019; Tao et al., 2016; Zhang et al., 2018), attributed to their N fixation capability (Vitousek et al., 2002). Leguminous shrubs also exhibit more stable N and P stoichiometry than non-leguminous shrubs (Guo et al., 2017; Zhang et al., 2018).

**4.3.3 Mycorrhizal associations**

Mycorrhizal type also impact the stoichiometry of shrubs. Chen et al. (2021) studied shrublands in peatlands in Northeast China and found that shrubs with ericoid mycorrhizae had higher C, C:N, C:P, and N:P ratios in their leaves, but lower N and P content, than those with ectomycorrhiza. Yang et al. (2021) analyzed the C, N, and P

stoichiometry of shrubs from 725 plots in northern China under different mycorrhizal types and reported that shrubs with ectomycorrhiza reflected higher P content in their leaves than those with arbuscular mycorrhizae, while N content did not differ significantly. This can be attributed to the fact that ectomycorrhizae have stronger phosphorus absorption than arbuscular mycorrhizae (Zhang et al., 2018; Toussaint et al., 2020).

**4.3.4 Shrub age**

Individual differences arising from different shrub ages can also influence leaf stoichiometry, yet there is no consensus on the direction of changes in leaf nutrient content with age. For instance, Zeng et al. (2017) studied the C, N, and P stoichiometry in leaves of *Caragana korshinskii* on the Loess Plateau and found that leaf C, N, and P content increased with shrub age, while C:N, C:P, and N:P ratios decreased. In contrast, Zhang et al. (2016) investigated the desert shrub *Haloxylon ammodendron* in North China and observed that leaf C and N content, as well as the N:P ratio, rapidly increased with stand age, while C:N significantly decreased, and P content and C:P ratio did not differ among age classes. Conversely, Dong et al. (2023) studied the evergreen shrub *Ammopiptanthus mongolicus* in the western Ordos Plateau and reported that with increasing shrub age, leaf N and P content decreased, while C:N, C:P, and N:P ratios increased, and C content remained relatively unchanged. ”

[Comment 6]: "6 Conclusion and Future Prospects": This section does not summarize and elaborate on the important conclusions of this study, such as how the stoichiometric characteristics of shrubs differ from those of forests and grasslands? The current research conclusion merely elaborates on the research results. It is suggested that the research results be further refined and summarized to draw important conclusions of this study.

[Response]: We have revised the Conclusion to further distill and synthesize the

study's findings. The revised text is as follows: "With the increasing recognition of the importance of shrublands and shrubs for ecosystem functioning, research into their stoichiometry has also increased, providing a more comprehensive understanding of the response and adaptive mechanisms of different vegetation types to climate change. This review highlights that, compared to forests and grasslands, shrubs exhibit distinct stoichiometric traits: leaf N and P concentrations are higher than those of trees but lower than those of herbs, while C content and C:N ratios fall between the two. Notably, the leaf N:P ratio in shrubs is generally higher than in other vegetation types, indicating a widespread P limitation in shrub ecosystems. These patterns reflect the unique nutrient-use strategies and adaptive mechanisms of shrubs under resource-limited conditions. Soil properties, climatic factors, and plant traits all influence shrub leaf stoichiometry, with leaf N and P content showing positive correlations with soil nutrient availability and precipitation, and negative correlations with temperature. Deciduous and leguminous shrubs tend to exhibit higher leaf N and P concentrations than evergreen and non-leguminous species. Increasing attention is being paid to the role of phylogenetic constraints in shaping stoichiometric patterns. Although substantial progress has been made in understanding the stoichiometric characteristics and driving factors of shrubs both domestically and internationally, several key scientific questions and research directions remain to be further explored."

[Comment 7]: "Future Prospects": The current research outlook mainly elaborates on the existing issues and deficiencies of the current research. It is necessary to further deepen the content of the research outlook and put forward innovative viewpoints to avoid vague expressions. It is suggested to adopt a three-part structure of "issue - method - value" for elaboration, and ultimately form an operational research path.

[Response]: We have revised the "Future Prospects" section to follow an issue－method－value structure and, in line with Reviewer 1's comments, added a discussion of methodological challenges that may affect comparability across studies. The

revised section is as follows:

**"6.1 Integrate Multi-organ and Plant–Soil–Microbe Stoichiometry in Shrub Ecosystems**

Current research on shrub stoichiometry is largely leaf-focused, with limited understanding of non-leaf organs such as stems, roots, reproductive organs, and litter, and of elemental coupling among organs. Integrated studies linking plants, soils, and microorganisms are also scarce, constraining insights into the mechanisms of C, N, and P cycling in shrub ecosystems. Future work should adopt coordinated multi-organ and aboveground–belowground sampling across climatic and edaphic gradients, combining standardized C, N, and P assays with techniques such as high-throughput sequencing, stable isotope tracing, and metabolomics to identify the stoichiometric profiles and functional roles of key microbial taxa. Analytical tools including linear regression and structural equation modeling (SEM) can then be used to reveal coordination and homeostatic mechanisms within plant–soil–microbe systems. This integrated approach will advance understanding of nutrient cycling in shrub ecosystems, refine ecological stoichiometry theory, and provide practical indicators for shrub restoration, grazing management, and enhancing ecosystem stability in arid regions.

**6.2 Consider Multi-factor Drivers and Successional Dynamics in Shrub Stoichiometry**

Shrub elemental composition is shaped by both phylogenetic constraints and multiple environmental drivers, yet most studies focus on single factors such as temperature, precipitation, and soil properties, with limited assessment of phylogenetic effects and little evaluation of atmospheric components (e.g., elevated $CO_2$, UV radiation, N deposition) and saline–alkaline stress or their interactions. In addition, shrub encroachment and shifts in dominant species can substantially alter community

stoichiometry, but the lack of long-term monitoring limits understanding of the underlying mechanisms. Future research should combine composite gradients of climate, soil, atmospheric chemistry, and salinity with temporal monitoring across encroachment stages, applying Bayesian phylogenetic mixed models and random forests models to partition the relative contributions and interactions of phylogeny and environment, alongside time-series and principal component analyses to identify patterns and drivers of elemental allocation. This approach will improve understanding of shrub adaptation under multiple stressors and succession, enhance predictions of nutrient dynamics under global change, and inform restoration and management of degraded ecosystems.

**6.3 Standardize Methodologies for Future Shrub Stoichiometry Research**

Differences in sampling organs, geographic sampling biases, statistical approaches, and uncertainties related to data extraction from published sources can all reduce the comparability of results across studies and affect their accuracy. For example, root elemental concentrations can vary substantially between fine roots, coarse roots, and roots of different diameter classes, making it difficult to compare findings across studies without consistent sampling protocols (You et al., 2023). Similarly, applying different analytical methods may lead to opposite conclusions about the relative importance of phylogeny and environment in shaping leaf stoichiometry, depending on the data structure and model assumptions (Tian et al., 2024). Future research should develop and adopt standardized protocols for sample collection, spatial representation, and data reporting, including clear organ classification, sample size requirements, geographic coverage, and analytical workflows, while assessing the robustness of results within a multi-model framework. Such measures will substantially improve the reliability, comparability, and integrative value of shrub stoichiometry research across regions and species, providing a solid empirical foundation for global-scale syntheses and model parameterization."

**Technical corrections:**

**[Comment 1]**: Line162-165: The number of decimal places retained in the text is inconsistent. Please unify them. It is recommended to check the full text and make revisions.

**[Response]:** Thank you for the helpful suggestion. We have standardized the number of decimal places in Lines 162-165 and conducted a thorough check of the entire manuscript. Please go to L164-167: "Among the 540 selected research articles in Chinese, 34.6% focused on the C, N, P stoichiometry of a specific organ in shrubs; most (80.75%) focused on leaves, a smaller proportion (9.10%) on roots, and yet another minority (8.56%) focused on litter, while only a few papers encompassed branches, bark, and seeds. Out of the 540 articles, 28.50% targeted shrubland soil, 25.90% addressed both shrubs and soil, and merely a 10.90% examined the stoichiometry across multiple shrub organs (Fig.2 inset)." And L396-400: "The global geometric means of C, N, and P content, as well as C:N and N:P ratios, in terrestrial plant leaves are 464.00 mg g$^{-1}$, 18.93 mg g$^{-1}$, 1.20 mg g$^{-1}$, 22.50, and 15.80, respectively. Among these, shrub leaves have mean values of 454.66 mg g$^{-1}$, 18.86 mg g$^{-1}$, 1.18 mg g$^{-1}$, 23.40, and 16.10, while trees show 502.31 mg g$^{-1}$, 16.58 mg g$^{-1}$, 1.08 mg g$^{-1}$, 30.10, and 15.40, and herbaceous plants 414.22 mg g$^{-1}$, 21.72 mg g$^{-1}$, 1.64 mg g$^{-1}$, 17.90, and 13.30 (Elser et al., 2000; Tian et al., 2018). In general, the N and P content in leaves follow the order of herbs > shrubs > trees, whereas the N:P ratio follows the reverse order." The decimal precision in Figure 3 has also been unified.

[Figure]

**Figure 3:** Changes in the number of papers published in the field of ecological stoichiometric studies of shrublands and shrubs from 1997 to 2023. The pie chart depicts the research subjects and their respective proportions in the literature on shrub ecological stoichiometry.

[Comment 2]: Line272: Why is there no linear regression analysis formula in Figure 4?

[Response]: We have added the linear regression equation to Figure 4 (now Figure 5), as shown below.

[Figure]

**Figure 5:** Relationship between leaf N, P and N:P ratio and annual mean precipitation and annual mean temperature of shrub leaves. The data are sourced from Tian et al. (2019), Li et al. (2021), Qin et al. (2022), Dong et al. (2023a), and Dong et al. (2023b).

[Comment 3]: Line408:Pay attention to the capitalization rules for word spelling in the title of the picture. For example, "Shrubland ecosystem" should be changed to "shrubland ecosystem".

[Response]: We have reviewed and corrected the capitalization in the figure titles; see

Lines 441: "**Figure 7:** Differences in leaf N: P stoichiometry of shrubs, trees and herbaceous. (a, b, c) shrubland ecosystem. (d, e, f) forest ecosystem."